# Laryngopharyngeal Reflux Patient Changes during the COVID-19 Quarantine

**DOI:** 10.3390/medicina59081475

**Published:** 2023-08-17

**Authors:** Alexandra Rodriguez, Younes Steffens, Christian Calvo-Henriquez, Miguel Mayo-Yáñez, Mihaela Horoi, Jerome R. Lechien

**Affiliations:** 1Department of Otolaryngology-Head and Neck Surgery, CHU Saint-Pierre, 1000 Brussels, Belgium; alexandra.rodriguez@stpierre-bru.be (A.R.); younes.steffens@stpierre-bru.be (Y.S.); mihaela.horoi@stpierre-bru.be (M.H.); 2Department of Otolaryngology-Head and Neck Surgery, Hospital Complex of Santiago de Compostela, 15706 Santiago de Compostela, Spain; chrisitan.calvo.henriquez@gmail.com; 3Otorhinolaryngology-Head and Neck Surgery Department, Complexo Hospitalario Universitario A Coruña (CHUAC), 15006 A Coruña, Spain; miguel.mayo.yanez@sergas.es; 4Department of Otolaryngology, Elsan Hospital, 86000 Paris, France; 5Department of Otolaryngology-Head and Neck Surgery, Foch Hospital, School of Medicine, University Paris Saclay, 75000 Paris, France; 6Department of Human Anatomy and Experimental Oncology, Faculty of Medicine, UMONS Research Institute for Health Sciences and Technology, University of Mons (UMons), 7000 Mons, Belgium

**Keywords:** reflux, laryngopharyngeal, larynx, laryngology, otolaryngology, head neck, gastroesophageal, lockdown, COVID-19, SARS-CoV-2, diet, stress, quarantine

## Abstract

*Background and Objective*: To examine the effects of the lockdown on diet adherence and stress levels in patients with laryngopharyngeal reflux (LPR). *Materials and Methods*: Patients with a positive LPR diagnosis at the hypopharyngeal-esophageal impedance-pH monitoring were treated from a pre- to lockdown period with a 3-month high-protein, low-fat, alkaline, plant-based diet, with behavioral changes, and an association of pantoprazole (20 MG/d) and alginate (Gaviscon 3/d). The following patient-reported outcomes questionnaire and findings instrument were used: Reflux Symptom Score-12 (RSS-12) and Reflux Sign Assessment (RSA). At the posttreatment time, patients were invited to evaluate the impact of lockdown on diet adherence and stress management with a predefined grid of foods and beverages and the perceived stress scale (PSS), respectively. *Results*: Thirty-two patients completed the evaluations. RSS-12 and RSA significantly improved from baseline to 3-month posttreatment. Most patients experienced mild-to-severe stress levels at the end of the lockdown. The level of stress substantially increased in 11 patients (34%) due to the lockdown, while it did not change in 11 patients (44%). In 11 cases (34%), patients reported that the adherence to the anti-reflux diet was better than initially presumed thanks to the lockdown period, while 44% (N = 14) reported that the lockdown did not impact the adherence to a diet. PSS and RSS-12 were significantly correlated at the end of the pandemic (r_s_ = 0.681; *p* < 0.001). The increase in stress level was positively associated with the lack of adherence to diet (r_s_ = 0.367; *p* = 0.039). *Conclusions*: During the lockdown, the diet habits of LPR patients were improved in one-third and unchanged in 44% of cases. The stress level was increased in one-third of patients, which was associated with an increase in symptom scores.

## 1. Introduction

Laryngopharyngeal reflux (LPR) is an inflammatory condition of the upper aerodigestive tract tissues related to the direct and indirect effects of gastroduodenal content reflux, which induces morphological changes in the upper aerodigestive tract [1]. LPR may concern 10% to 30% of outpatients consulting in the otolaryngology department [1]. The most common symptoms include throat clearing, globus sensation, throat pain, and cough, while gastroesophageal reflux disease (GERD) typical symptoms, such as heartburn or regurgitation, are often lacking [1,2]. The consumption of high-fat, high-quick-release sugar, and low-protein foods and beverages and stress (autonomic nerve dysfunction) are both factors that may negatively influence the esophageal sphincter tonicity, leading to pharyngeal reflux events [1,2]. It has been suggested that high-fat, high- quick-release sugar and low-protein foods may reduce the lower and upper esophageal sphincter tonicities, increase the numbers of transient sphincter relaxation and related gaseous reflux events, last the gastric emptying time, and reduce the up to down motility of esophagus [1]. Regarding autonomic nerve function, LPR was associated with an imbalance between sympathetic and para-sympathetic nerve functions, decreasing the para-sympathetic function, which was associated with esophageal sphincter and body dysfunction [1,2].

With the recent coronavirus disease 2019 (COVID-19) pandemic, many countries imposed lockdowns to reduce the virus spread in the population. Many citizens were confined to home for several weeks, which should influence positively [3] or negatively [4] individual lifestyles and diet habits. Indeed, some patients decided to improve their cooking habits with natural and healthy products, while others increased their consumption of alcohol, snacks, and fast-food to decrease the stress related to the pandemic [3,4].

The objective of this study was to examine the effects of the COVID-19 lockdown on diet adherence and stress levels in patients undergoing treatment for laryngopharyngeal reflux (LPR).

## 2. Methods

Patients with laryngopharyngeal symptoms, findings, and a positive LPR diagnosis at the 24-h hypopharyngeal-esophageal multichannel intraluminal impedance pH-monitoring (HEMII-pH) prior to the COVID-19 lockdown were followed throughout the lockdown period (March to December 2020) in the Department of Otolaryngology–Head and Neck Surgery of CHU Saint-Pierre (Brussels, Belgium). The LPR diagnosis was based on the occurrence of LPR symptoms and >1 acid, weakly acid, or nonacid pharyngeal reflux events at the HEMII-pH (OFF medication) [5]. To study the influence of lockdown on diet adherence and stress management, we only included patients who started the treatment just before the lockdown, while they were followed throughout the lockdown periods. Patients with another source of stress during the lockdown (other than the pandemic) or those who did not adhere to the anti-reflux diet were excluded.

The local ethics committee approved the study protocol (CHUSP, n°BE076201837630). Patients consented to participate.

### 2.1. Hypopharyngeal-Esophageal Multichannel Intraluminal Impedance-pH Testing

The probe placement and configuration characteristics were detailed in previous publications [6].

Briefly, the catheter was placed in the morning before breakfast (8:00 AM) and removed the next day in the morning. The catheter was composed of 8 impedance segments and 2 pH electrodes (Versaflex Z^®^, Digitrapper pH-Z testing System, Medtronic, Dublin, Ireland, Europe). The six esophageal impedance segments were placed along the esophagus zones (Z1 to Z6) at 19, 17, 11, 9, 7, and 5 cm above the lower esophageal sphincter (LES). The pharyngeal impedance segments were placed 1 and 2 cm above the upper esophageal sphincter (UES) in the hypopharynx. The pH electrodes were placed 2 cm above LES and 1- to −2 cm below UES, respectively. According to a recent systematic review providing normative data for HEMII-pH, the LPR diagnosis criteria were based on the occurrence of >1 acid (pH ≤ 4.0), weakly acid (pH = 4.0–7.0), or nonacid (pH > 7.0) hypopharyngeal reflux events (off proton pump inhibitors) [5].

### 2.2. Clinical and Therapeutic Outcomes

Symptoms were evaluated with reflux symptom score-12 (RSS-12) [7], which is a validated 12-item patient reported-outcome questionnaire including otolaryngological, digestive, and respiratory symptoms of reflux (Figure 1). The total score ranged from 0 (no symptom) to 300 (frequent and severe symptoms). Reflux sign assessment (RSA) is a validated 61-item clinical instrument, which was developed to rate oral, pharyngeal, and laryngeal findings associated with LPR throughout the treatment period [8]. RSA rates the signs associated with LPR, such as laryngeal erythema or edema (Figure 2). The stress level of the patient was evaluated at the end of the lockdown with the Perceived Stress Scale (PSS), which is a 10-item validated patient-reported outcome questionnaire [9]. The normative data reported that a PSS < 12 was normal [9]. A PSS score between 12 and 21 consists of patients who have stress but are adequately managed (mild stress). The stress is moderately managed when the score ranges from 21 to 26 (moderate stress). PSS >26 corresponds to an inadequately managed stress (severe stress) [9].

### 2.3. Treatment

According to the HEMII-pH findings of reflux, patients benefited from a 3-month treatment combining anti-reflux diet, proton pump inhibitors (PPIs; pantoprazole 20 mg once daily), alginate (Gaviscon^®^ 3/d, Reckitt Benckiser, Slough, UK) or magaldrate (Riopan^®^ 3/d, Takeda, Zaventem, Belgium) [6]. Patients with acid reflux benefited from pantoprazole (20 MG, once daily, fasting) and post-meal alginate (thrice daily), while those with nonacid reflux were treated with post-meal magaldrate or alginate only. Individuals with weak acid reflux received a combination of pantoprazole (20 MG, once daily, fasting) and thrice daily post-meal alginate or magaldrate. Patients with nighttime reflux at the HEMII-pH tracing benefited from additional alginate or magaldrate (alkaline LPR) at bedtime [6].

At the first consultation, patients were invited to specify ‘refluxogenic’ foods and beverages that they commonly consumed through a predefined list [10]. The anti-reflux diet was standardized [10], and based on the reduction of foods and beverages associated with a high risk of reflux [10], and the consumption of high-protein, low-fat, alkaline, plant-based foods and beverages [6,10]. The list of foods and beverages, which are recommended or discouraged according to these documented impacts on gastroesophageal function is provided in Figure 1 and Figure 2. At 3-month posttreatment, patients were invited to specify which foods and beverages they succeeded in decreasing or stopping. The medications were titrated at this time regarding the 3-month RSS-12 considering a reduction of >20% of the baseline RSS-12 as an adequate therapeutic response.

### 2.4. Lockdown Evaluations

Patients were invited to evaluate the influence of the lockdown on both diet adherence and stress level through a short patient-reported outcome questionnaire at baseline and at 3-month posttreatment.

### 2.5. Statistical Analyses

Statistical analyses were performed with the Statistical Package for the Social Sciences for Windows (SPSS version 24.0; IBM Corp, Armonk, NY, USA). Wilcoxon rank test was used to evaluate the evolution of RSS-12, RSA, and PSS from baseline to 3-month posttreatment. The consumption of foods and beverages (weekly versus daily versus no) was assessed with chi-square. There was no sample size calculation. Spearman analysis was performed to assess the relationship between outcomes. A *p*-value < 0.05 was considered as significant.

## 3. Results

Thirty-two patients met the inclusion criteria and completed the evaluations. The mean age of patients was 50.5 ± 16.4 years. There were 22 females (69%) and 10 males (31%). The clinical features of patients are described in Table 1. The mean RSS-12 significantly improved from baseline (66.6 ± 49.1) to 3-month posttreatment (47.6 ± 39.2; *p* = 0.008). The mean pre-treatment RSA (24.2 ± 11.2) significantly improved at 3-month posttreatment (20.3 ± 9.5; *p* = 0.031). Twenty-five patients (78.1%) reported significant symptom reduction (>20% reduction of baseline RSS-12) at the posttreatment time and were considered as responders (Table 1).

### Influence of Lockdown

The pre- to post-lockdown evolution of patient consumption of ‘refluxogenic’ foods and beverages is reported in Table 2. According to the Wilcoxon rank test, patients significantly decreased most foods and beverages associated with a high risk of reflux event. Eleven patients (34.4%) reported that the adherence to the anti-reflux diet was better than initially presumed thanks to the lockdown period, while 14 (43.8%) believed that the lockdown did not impact the adherence to diet (Figure 3A).

The mean PSS at the end of the lockdown was 28.3 ± 8.8, which corresponded to high-stress levels regarding normative data (threshold = 12.8 ± 6.2) [8]. The stress was related to the pandemic and lockdown. Precisely, only one patient reported PSS < 13 (3.1%). According to the PSS, 5 (15.6%), 6 (18.8%), and 20 (62.5%) patients reported mild, moderate and severe stress, respectively. Three patients (9.4%) thought that the lockdown period was associated with a better decrease in stress than initially presumed thanks to the lockdown period. Eleven patients (34.4%) believed that the lockdown period increased their stress level, while there was no influence of the lockdown on stress in 18 patients (56.3%; Figure 3B).

Overall, 6 patients (18.8%) reported that the lockdown had a negative impact on their LPR. The PSS and RSS-12 scores at the end of the lockdown were significantly correlated (r_s_ = 0.681; *p* < 0.001). There was a positive association between the stress increase and the lack of adherence to diet at the end of the pandemic (r_s_ = 0.367; *p* = 0.039).

## 4. Discussion

The success of LPR treatment depends on many factors, such as adherence to low-acid, low-fat, and high-protein diets and the management of stress and related autonomic nerve dysfunction [10]. Many countries have forced the quarantine of some regions to limit the spread of the virus [11,12,13], which has confined citizens at home for several weeks. Regarding the high prevalence of LPR in the population [14,15] and the potential impact of lockdown on diet habits and stress, we aimed to investigate how the lockdown has influenced the therapeutic outcomes of LPR patients.

In the present study, we observed that LPR patients who were diagnosed just before the lockdown period adhered adequately to the anti-reflux diet, and mainly reported favorable or neutral influence of lockdown on their diet adherence. The positive or neutral impact of lockdown on the diet habits of patients corroborated the findings of some previous studies [3,16], while others reported mitigated impact of lockdown on diet habits [4]. In a recent meta-analysis of 42 studies, Della Valle et al. reported that 85% of studies measuring changes in Mediterranean diet adherence before versus during lockdown reported an increased rate of change of high adherence to diet, which ranged from +3.3% to +21.9% of cases [3]. Similar findings were observed by Alverez-Gomez et al. who found that the quarantine period was associated with a better, healthy lifestyle and dietary habits of the Spanish population compared to the pre-quarantine period. Precisely, they reported high consumption of fruits, vegetables, and legumes, as well as adequate time to prepare meals [16]. The pre- to posttreatment specific analysis of diet changes in the present study supports that patients have decreased high-fat, high-quick-release sugar, and refluxogenic foods and beverages, which was associated with LPR symptom relief or significant reduction [11].

The pandemic situation may be associated with an increase in stress, anxiety, and autonomic nerve dysfunction [17,18]. Autonomic nerve dysfunction may be characterized by a reduction of the vagus nerve activity on the esophagus body and sphincters [19], which may be associated with an increased number of transient lower and upper esophageal sphincter relaxation episodes, and consequently, the backflow of gastric content into the upper aerodigestive tract. In LPR, pharyngeal reflux events are mainly gaseous and occur post-meals or at a distance of the meals [20].

In LPR disease, it has been supported that the patients with stress and anxiety had impaired autonomic nerve function with higher heart rate variability than controls [2,19]. In that way, Wang et al. reported that patients with anxiety or stress may have more severe LPR symptoms compared with those without significant autonomic nerve dysfunction. In the present study, patients reported mild-to-severe stress levels and 34% of patients reported a negative impact of lockdown on stress. Moreover, 56% did not report lockdown influence. Interestingly, the stress score was significantly associated with RSS-12 at the end of the lockdown, which supports the influence of stress on LPR disease. The findings of the present study were particularly important regarding the potential increase of LPR symptoms in COVID-19 [21], which may be attributed to the increase of both anxiety and stress in the population. In that way, a recent meta-analysis suggested an increase in dysphonia during and post-COVID-19 infection [22], which should be attributed to LPR. The occurrence of chronic vagus nerve dysfunction in long COVID-19 patients [23] is an additional important issue to explore in future studies. The evolution of stress and anxiety on LPR during lockdown is an important issue according to the burden and cost related to the management of LPR and related complications (e.g., upper aerodigestive tract infections, nasal inflammatory conditions, otitis media, increased risk of vocal cord dysfunction, etc.) [1]. The study results may support the need for prevention in potential future similar situations.

The anxiety and stress were commonly attributed to the pandemic and lockdown. Future studies should be interested in determining the factors underlying stress and anxiety, which may include social isolation, economic concerns, fear of dying, or of the unknown, etc. [24,25].

However, the study has some limitations. The low number of patients and the lack of objective testing of autonomic nerve dysfunction (e.g., heart rate variability device) were the primary limitations. The lack of calculation of effect size is an additional limitation. However, it was difficult to include more patients regarding the short and unpredictable period of study (lockdown periods) and the need to include patients with an objective LPR diagnosis (HEMII-pH). The lack of evaluation of stress during the pre-lockdown period is an additional limitation. The low number of patients may make the findings of the study not generalizable, while it is possible that the level of stress prior to the lockdown was under- or overestimated by patients who evaluated the evolution during the lockdown after potential habituation. The use of HEMII-pH to confirm the diagnostic and the use of validated reflux and stress scales were the main strengths of the study. Indeed, the LPR diagnostic is complicated with non-specific symptoms and signs [21], and the related risk of false positive.

## 5. Conclusions

To the best of our knowledge, this is the first study describing the lockdown influence on diet habits and stress of patients with LPR disease. The diet habits were improved or unchanged in most cases, while stress level was increased in one-third of patients. Patients with a high level of stress related to the pandemic/lockdown situation reported high reflux symptom scores. The management of stress during the lockdown and pandemic periods is an important issue in LPR patients and needs future prospective controlled studies.

## Figures and Tables

**Figure 1 medicina-59-01475-f001:**
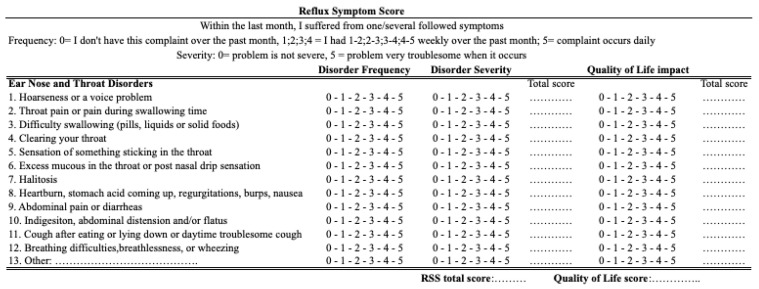
Reflux Symptom Score-12. footnotes: Reflux Symptom Score-12 is a validated patient-reported outcome questionnaire assessing LPR symptoms. The severity item (5-point) is multiplied by frequency (5-point) to obtain a symptom score (0–25). The sum is calculated to obtain RSS-12 final score (0–300).

**Figure 2 medicina-59-01475-f002:**
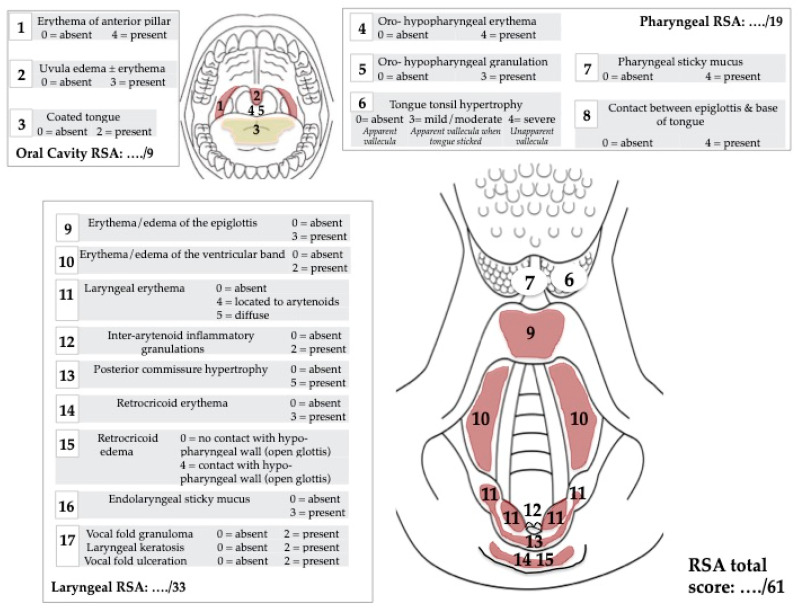
Reflux Sign Assessment. footnotes: Reflux Sign Assessment is a clinical instrument assessing oral, pharyngeal, and laryngeal findings associated with LPR.

**Figure 3 medicina-59-01475-f003:**
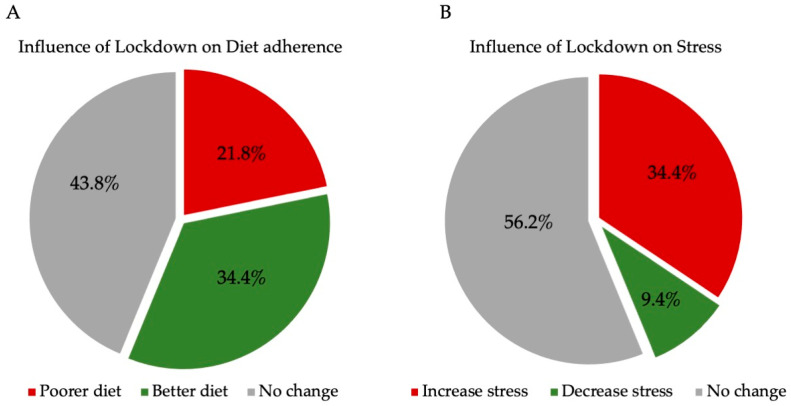
Evolution of Stress management and Diet adherence in lockdown period. The figure highlights the influence of lockdown on diet adherence and stress levels according to the patient evaluation.

**Table 1 medicina-59-01475-t001:** Cohort characteristics.

Characteristics	N = 32 Patients
**Mean age (SD)**	51.8 ± 17.7
**Age groups (N, %)**	
18–30 years	3 (9)
31–50 years	7 (22)
>51 years	22 (69)
**Body mass index**	25.1 ± 4.7
**Gender (N, %)**	
Male	10 (31.3)
Female	22 (68.7)
**Gastrointestinal endoscopy**	N = 21
Normal	2 (9.5)
Hiatal hernia	7 (33.3)
LES insufficiency	13 (61.9)
Esophagitis	11 (52.4)
Gastritis	10 (47.6)
Helicobacter Pylori infection	1 (4.8)
**Types of LPR at the HEMII-pH**	
Acid LPR	15
Weakly acid LPR	10
Nonacid LPR	7
**HEMII-pH feature** (m ± SD)	
Pharyngeal acid reflux episodes	34.8 ± 36.1
Pharyngeal nonacid reflux episodes	24.4 ± 18.3
Pharyngeal reflux episodes upright	22.1 ± 15.8
Pharyngeal reflux episodes supine	3.7 ± 5.3
Pharyngeal reflux episodes (total)	57.3 ± 44.2
**GERD**	17 (51.5)
Percentage of time with distal pH < 4	6.9 ± 14.5
DeMeester score	20.4 ± 41.7
**Responder rates**	N (%)
No response (chronic course)	7 (21.9)
Mild response	2 (6.2)
Moderate response	7 (21.9)
High response	12 (37.5)
Complete responses	4 (12.5)

Abbreviations: GERD = gastroesophageal reflux disease; HEMII-pH = hypopharyngeal-esophageal multichannel intraluminal impedance-pH monitoring; LES = lower esophageal sphincter; LPR = laryngopharyngeal reflux.

**Table 2 medicina-59-01475-t002:** Baseline and pandemic diet habits of patients.

	Pre-Treatment			Pandemic			
Refluxogenic Diet Outcomes	Weekly	Daily	Tot (%)	Weekly	Daily	Tot (%)	*p*-Value
Fat fish, fish oil (sardines, cods, herrings)	23	0	23 (71.9)	13	0	13 (40.6)	0.003
Fat chicken	17	0	17 (53.1)	1	0	1 (3.1)	NS
High-fat meat							
Kidney	5	0	5 (15.6)	13	0	13 (40.6)	NS
Sheep meat	13	0	13 (40.6)	3	0	3 (9.4)	NS
Lamb meat	24	0	24 (75.0)	17	0	17 (53.1)	0.001
Bacon	17	0	17 (53.1)	7	0	7 (21.9)	NS
Beef meat	25	0	25 (78.1)	18	0	18 (56.3)	0.001
Porc meat	18	0	18 (56.3)	10	0	10 (31.3)	0.018
Ground	30	0	30 (93.8)	25	0	25 (78.1)	0.001
Pate	13	0	13 (40.6)	1	0	1 (3.1)	NS
Tripe	4	0	4 (12.5)	12	0	12 (37.5)	NS
Charcuterie	19	5	24 (75.0)	8	3	11 (34.4)	0.005
Chocolate	20	8	28 (87.5)	9	4	13 (40.6)	0.001
Chocolate cookies	20	6	26 (81.3)	9	3	12 (37.5)	0.001
Full-fat cheese	19	10	29 (90.6)	14	5	19 (59.4)	0.001
Whole milk	10	2	12 (37.5)	14	0	14 (43.8)	NS
Ice cream	25	1	26 (81.3)	15	0	15 (46.9)	0.001
Peanut, nut, cashew, hazelnut	23	1	24 (75.0)	11	0	11 (34.4)	0.007
French fries & frying	28	1	29 (90.6)	18	0	18 (56.3)	0.001
Shallot or onion	21	6	27 (84.4)	16	3	19 (59.4)	0.001
Spicy	16	15	31 (96.9)	17	8	25 (78.1)	0.001
Chilli	16	0	16 (50.0)	1	0	1 (3.1)	0.001
Tomato (sauce or raw tomato)	28	3	31 (96.9)	20	1	21 (65.6)	0.001
Strong alcohols	12	1	13 (40.6)	14	1	15 (46.9)	NS
Wines	15	8	23 (71.9)	10	4	14 (43.8)	0.001
Beer	12	4	16 (50.0)	6	2	8 (25.0)	0.014
Sparkling beverage (water, soda)	19	2	21 (65.6)	6	0	6 (18.8)	NS
Coffee	8	17	25 (78.1)	8	7	15 (46.9)	0.001
Tea	14	12	26 (81.3)	12	6	18 (56.3)	0.001
Orange, grapefruit or high-sugar juices	23	1	24 (75.0)	10	0	10 (31.3)	0.008
Sauces (mayonnaise, mustard, ketchup, etc.)	30	1	31 (96.9)	20	0	20 (62.5)	0.001
Bakery	28	1	29 (90.6)	19	0	19 (59.4)	0.001
Sirup	9	2	11 (34.4)	2	0	2 (6.3)	NS
Butter products	16	12	28 (87.5)	10	8	18 (56.3)	0.001
Sweets	18	0	18 (56.3)	2	0	2 (6.3)	NS

The weekly and daily consumption of refluxogenic foods and beverages were reported in both the pre-treatment and pandemic periods. For each period, the number of patients reporting consuming weakly and daily foods or beverages were reported, as well as the sum (N (%)) of weekly and daily refluxogenic foods and beverages. According to Chi-square, the consumption of most refluxogenic foods and beverages significantly decreased from pre-treatment to the lockdown period.

## Data Availability

The data are available on request.

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
