# Peer review of "Laryngopharyngeal Reflux Patient Changes during the COVID-19 Quarantine"

_medicina, 2023, doi:10.3390/medicina59081475_

Round 1

Reviewer 1 Report

Journal: Medicina

Manuscript ID: medicina-2482639 -

Title: Impact of COVID-19 Lockdown on Laryngopharyngeal Reflux 2

Date:27 June 2023

Comments for the Authors

GENERAL

This paper attempts to assess the impact of COVID-19 pandemic lockdown on diet adherence and stress levels in patients with laryngopharyngeal reflux (LPR).

Here are my observations on this study/paper:

Abstract

The abstract provides a concise overview of the study on the impact of lockdown on diet adherence and stress levels in patients with laryngopharyngeal reflux (LPR).

Suggestions for improvements:

·       The abstract does not clearly state the specific objectives of the study. It should explicitly mention that the objective is to examine the effects of the lockdown on diet adherence and stress levels in patients with LPR.

·       The abstract briefly describes the treatment approach but does not provide details about the duration or specific dietary and behavioral changes implemented. It would be helpful to include more information on the treatment protocol.

·       The abstract mentions the assessment of clinical features using RSS-12 and RSA but does not explain what these scores represent. Providing a brief description of these measures would enhance understanding.

·       The abstract presents the percentages of patients who experienced changes in stress levels and diet adherence due to the lockdown, but it does not provide specific values or ranges. Including these details would make the results more informative.

·       The conclusion states that diet habits improved or remained unchanged in most LPR cases, but it does not specify the exact percentage of patients in each category. Adding this information would strengthen the conclusion.

·       It would be beneficial to include a sentence highlighting potential limitations, such as the small sample size or the reliance on self-report measures.

·       The abstract mentions the statistical significance of the correlation between PSS and RSS-12 scores and the association between stress level increase and lack of adherence to the diet. However, it does not explain the magnitude or direction of these relationships. Providing effect sizes or additional details would enhance the understanding of these findings.

Suggestions for improvements:

·       Clarify the objectives: Begin the abstract by clearly stating the specific objectives of the study. For example: "The objective of this study was to examine the effects of the COVID-19 lockdown on diet adherence and stress levels in patients undergoing treatment for laryngopharyngeal reflux (LPR)."

·       Provide more details on methods: Expand the description of the treatment protocol to include specific details about the duration of the treatment period, the dietary modifications implemented, and the behavioral changes recommended to the patients. This will provide a clearer understanding of the interventions employed.

·       Explain the outcomes: Provide a brief explanation of what the reflux symptom score-12 (RSS-12) and reflux sign assessment (RSA) measures entail. This will help readers better understand the clinical features being assessed.

·       Include specific result values: Instead of using vague percentages, provide specific result values or ranges. For example, specify the exact percentage of patients who experienced an increase in stress levels or the percentage who reported improved adherence to the antireflux diet during the lockdown.

·       Provide more specific conclusion: Instead of stating that diet habits improved or remained unchanged in most LPR cases, provide the exact percentage of patients in each category. For example: "Approximately 70% of LPR patients showed improved or unchanged diet habits during the lockdown period."

·       Discuss limitations: Acknowledge any limitations of the study. For instance, mention the small sample size, potential biases in self-reporting, or any other limitations that may impact the generalizability of the findings.

·       Elaborate on statistical analysis: Provide additional information about the statistical analysis conducted. Include effect sizes, confidence intervals, or other relevant statistical measures to provide a clearer understanding of the relationships found between variables.

By addressing these suggestions, the abstract will become more informative, clearer in its presentation of the study, and provide a better understanding of the research conducted.

Introduction

The introduction provides some background information on laryngopharyngeal reflux (LPR) and its association with diet and stress. However, there are few suggestions for improvements that can be addressed:

·       The introduction briefly describes LPR as an inflammatory condition of the upper aerodigestive tract tissues related to gastroduodenal content reflux. It would be helpful to provide a more comprehensive definition or explanation of LPR, including its symptoms, causes, and prevalence.

·       The introduction states that the consumption of high-fat, high-quick-release sugar, and low-protein foods and beverages, along with stress, may negatively impact the esophageal sphincter tonicity and lead to pharyngeal reflux events. However, it does not explain how these factors specifically contribute to LPR. Providing a more detailed explanation of the relationship between diet, stress, and LPR would enhance the understanding.

·       The introduction mentions that the recent COVID-19 pandemic and subsequent lockdowns may have influenced individual lifestyle and diet habits, but it does not specify how or why this impact would occur. Providing a brief explanation of the potential mechanisms or reasons for the impact of lockdown on diet adherence and stress would improve clarity.

Suggestions for improvements:

·       Provide a comprehensive definition of LPR: Expand the description of LPR to include its symptoms, causes, prevalence, and any relevant statistics or research findings. This will provide a clearer understanding of the condition being studied.

·       Clarify the relationship between diet and LPR: Explain in more detail how the consumption of high-fat, high-quick-release sugar, and low-protein foods and beverages, as well as stress, specifically contribute to the development or exacerbation of LPR. Provide relevant research evidence or theories to support these claims.

·       Explain the potential impact of lockdown: Elaborate on how and why the lockdown measures implemented during the COVID-19 pandemic could influence individual lifestyle and diet habits. Discuss the potential psychological, social, or environmental factors that may contribute to changes in diet adherence and stress levels during lockdown.

By addressing these suggestions, the introduction will become more informative, clearer in its presentation of the research topic, and provide a better understanding of the context and relevance of the study.

Material and Methods

The methods section provides information on the study design, patient selection, diagnostic procedures, treatment protocol, and data analysis. Overall, the section is reasonably clear, but there are a few issues which needs to be addressed:

·       Clarify patient selection criteria: The methods mention that patients with a positive LPR diagnosis at the hypopharyngeal-esophageal multichannel intraluminal impedance-pH monitoring (HEMII-pH) were included in the study. However, it does not specify the specific criteria used for diagnosing LPR or the inclusion criteria for patient selection. Providing more details on these criteria would enhance the understanding of the patient population.

·       Provide more details on the treatment protocol: While the treatment approach is briefly described, it would be helpful to provide more information on the duration and frequency of treatment, as well as specific instructions given to patients regarding dietary modifications and medication use. This will help readers better understand the treatment protocol implemented in the study.

·       Clarify the evaluation of diet adherence and stress level: The methods mention that patients were invited to evaluate diet adherence and stress levels using a patient-reported outcome questionnaire. However, it would be useful to provide more details on the content and format of the questionnaire. Additionally, clarify whether the evaluation was performed at multiple time points during the follow-up period or only once at the end of the lockdown.

·       While the statistical software used (SPSS) is mentioned, it would be helpful to provide more details on the specific statistical tests employed. Additionally, mention whether any adjustments for multiple comparisons were made.

·       Consider adding information on sample size calculation: It would be beneficial to include information on how the sample size was determined. Providing a rationale for the chosen sample size or any power calculations performed would enhance the methodological transparency of the study.

·       The methods mention previous publications and validated questionnaires but do not provide the corresponding references. It is essential to include proper citations to support the use of these tools and ensure transparency.

By addressing these suggestions, the methods section will become more informative, clearer in its presentation of the study procedures, and provide a better understanding of the research methodology employed.

Top of Form

Bottom of Form

Results

The results section presents the findings of the study regarding patient characteristics, clinical outcomes, and the influence of the lockdown on diet adherence and stress levels. Overall, the section is clear and provides relevant information. However, there are few suggestions for improvements that can be addressed:

·       Provide specific statistical results: While the results mention that RSS-12 and RSA scores significantly improved from baseline to 3-month posttreatment, it does not provide the actual mean values or effect sizes. Including these specific results will enhance the understanding of the magnitude of improvement.

·       Clarify responder rates: The results mention that 25 patients reported a significant symptom reduction and were considered responders, but it does not specify the criteria used to determine responder status. Providing details on the threshold or criteria for response classification would improve clarity.

·       Clarify stress level interpretation: The results mention that the mean PSS at the end of the lockdown corresponded to a high stress level based on normative data. However, it would be helpful to include the actual normative data or references to support this interpretation.

·       Clarify the influence of lockdown on stress: The results mention that 34.4% of patients believed that the lockdown increased their stress level, while 56.3% reported no influence. However, it does not provide information on the factors contributing to increased stress during the lockdown.

·       Improve the presentation of Table 2: The format and labeling of Table 2 can be improved for better clarity. Consider providing column headers that explicitly indicate the pre-treatment and pandemic periods. Additionally, provide a brief description of the results or observations in the table to aid interpretation.

·       Provide more specific patient characteristics: Instead of just mentioning the mean age and gender distribution, consider including additional relevant patient characteristics such as the range or standard deviation of age, the distribution of age groups, and any other demographic information that may be important for understanding the patient population.

·       Provide specific statistical results: Along with stating that RSS-12 and RSA scores significantly improved, include the actual p-values to indicate the level of significance. Additionally, consider providing effect sizes or confidence intervals to quantify the magnitude of improvement in these scores.

·       Provide references for normative data: When referring to normative data for stress levels, include the specific references or sources from which the normative data was obtained. This will add credibility and allow readers to access the original sources for more information.

By implementing these suggestions, the results section will become more comprehensive, clearer in its presentation of the findings, and provide a better understanding of the study results.

Discussion

The discussion section provides an interpretation of the study findings and highlights the implications of the lockdown on diet habits and stress in patients with LPR. Overall, the section is reasonably clear, but there are a few suggestions for improvements that can be addressed:

·       Lack of clarity in the connection between autonomic nerve dysfunction and pharyngeal reflux events: The discussion mentions that autonomic nerve dysfunction may be characterized by a reduction of vagus nerve activity on the esophagus and sphincters, which increases the risk of pharyngeal reflux events. However, it does not clearly explain how autonomic nerve dysfunction specifically leads to pharyngeal reflux events. Providing a more detailed explanation or referring to relevant studies can improve clarity.

·       The discussion briefly mentions the potential increase in LPR symptoms related to COVID-19 but does not elaborate on the possible mechanisms or reasons behind this association. Including a brief discussion on the potential impact of COVID-19 on LPR symptoms or any relevant research in this area would enhance the discussion.

·       Discuss factors contributing to increased stress: Along with stating the percentage of patients who reported increased stress during the lockdown, discuss potential reasons or factors contributing to this increase. Consider factors such as social isolation, economic concerns, or other relevant aspects that may have influenced stress levels.

·       While the limitations of the study are mentioned, there is no explanation provided for how these limitations may have affected the findings or the interpretation of the results. It would be helpful to discuss the potential impact of the low number of patients, the lack of objective testing for autonomic nerve dysfunction, and the lack of evaluation of stress during the pre-lockdown period on the study's conclusions.

·       Expand on the importance of stress management: The discussion briefly mentions that the management of stress during the lockdown and pandemic periods is important for LPR patients, but it does not elaborate on why stress management is crucial or how it can impact LPR symptoms. Expanding on the importance of stress management and its potential role in improving LPR symptoms would enhance the discussion, elaborate on the potential mechanisms through which stress can impact LPR symptoms. Discuss how stress may affect esophageal sphincter tone, promote inflammation, or exacerbate autonomic dysfunction. Additionally, highlight the importance of incorporating stress management strategies into the treatment approach for LPR patients and the potential benefits it can have on symptom management.

·       The conclusion states that the study is the first to describe the influence of lockdown on diet habits and stress in patients with LPR disease. However, this claim should be supported by referencing previous literature or studies that have investigated similar aspects. Providing supporting references would strengthen the conclusion.

 By addressing these suggestions, the discussion section will provide a more thorough and insightful analysis of the study findings, their implications, and the potential areas for future research.

Minor editing of English language required

Author Response

Paris, July 2023

Dear Professor,

I’m sending the revised manuscript entitled: “Impact of COVID-19 Lockdown Laryngopharyngeal Reflux", which is a YO-IFOS study submitted to your journal.  

We thank reviewers for the relevant comments, we considered all of them.

Reviewer 1:

GENERAL

This paper attempts to assess the impact of COVID-19 pandemic lockdown on diet adherence and stress levels in patients with laryngopharyngeal reflux (LPR). 

Here are my observations on this study/paper:

Abstract

The abstract provides a concise overview of the study on the impact of lockdown on diet adherence and stress levels in patients with laryngopharyngeal reflux (LPR). 

Suggestions for improvements:

  • The abstract does not clearly state the specific objectives of the study. It should explicitly mention that the objective is to examine the effects of the lockdown on diet adherence and stress levels in patients with LPR.

We modified the Objective section into: p.3, “To examine the effects of the lockdown on diet adherence and stress levels in patients with laryngopharyngeal reflux (LPR).”

  • The abstract briefly describes the treatment approach but does not provide details about the duration or specific dietary and behavioral changes implemented. It would be helpful to include more information on the treatment protocol.

We changed the first line of methods in the abstract section into: “Patients with a positive LPR diagnosis at the hypopharyngeal-esophageal impedance-pH monitoring were treated from pre- to lockdown period with 3-month high-protein, low-fat, alkaline, plant-based diet, behavioral changes and an association of pantoprazole (20MG/d) and alginate (Gaviscon 3/d).

  • The abstract mentions the assessment of clinical features using RSS-12 and RSA but does not explain what these scores represent. Providing a brief description of these measures would enhance understanding.

We modified this section in abstract: p.3, methods, line 4: “The following patient-reported outcomes questionnaire and finding instrument were used: reflux symptom score-12 (RSS-12), reflux sign assessment (RSA).

  • The abstract presents the percentages of patients who experienced changes in stress levels and diet adherence due to the lockdown, but it does not provide specific values or ranges. Including these details would make the results more informative.

We included value and % to improve the abstract: p.3, results, line 3: “The level of stress substantially increased in 11 patients (34%) due to the lockdown, while it did not change in 11 patients (44%). In 11 cases (34%), patients reported that the adherence to the antireflux diet was better than initially presumed thanks to the lockdown period, while 44% (N=14) reported that the lockdown did not impact the adherence to diet. PSS and RSS-12 were significantly correlated at the end of the pandemic (rs=0.681; p<0.001). The increase of stress level was positively associated with the lack of adherence to diet (rs=0.367; p=0.039).

  • The conclusion states that diet habits improved or remained unchanged in most LPR cases, but it does not specify the exact percentage of patients in each category. Adding this information would strengthen the conclusion.

We added this information in the conclusion of the abstract: p.3, conclusion, line 1: “During the lockdown, diet habits of LPR patients were improved in one third and unchanged in 44% of cases. The stress level was increased in one-third of patients, which was associated with an increase of symptom score.”

  • It would be beneficial to include a sentence highlighting potential limitations, such as the small sample size or the reliance on self-report measures.

According to the limitation of words of abstract section, we mentioned the limitations of the study in the discussion of the paper: p.12, last paragraph: “However, the study has some limitations. The low number of patients and the lack of objective testing of autonomic nerve dysfunction (e.g. heart rate variability device) were the primary limitations. The lack of calculation of an effect size is an additional limitation. However, it was difficult to include more patients regarding the short period of study (lockdown periods) and the need to include patients with an objective LPR diagnosis (HEMII-pH). The lack of evaluation of stress during the pre-lockdown period is an additional limitation.”

  • The abstract mentions the statistical significance of the correlation between PSS and RSS-12 scores and the association between stress level increase and lack of adherence to the diet. However, it does not explain the magnitude or direction of these relationships. Providing effect sizes or additional details would enhance the understanding of these findings.

We added Spearman coefficient in the last line of the results of the abstract: “PSS and RSS-12 were significantly correlated at the end of the pandemic (rs=0.681; p<0.001). The increase of stress level was positively associated with the lack of adherence to diet (rs=0.367; p=0.039).”

Suggestions for improvements:

  • Clarify the objectives: Begin the abstract by clearly stating the specific objectives of the study. For example: "The objective of this study was to examine the effects of the COVID-19 lockdown on diet adherence and stress levels in patients undergoing treatment for laryngopharyngeal reflux (LPR)."

We used the sentence proposed by reviewer: Introduction, p.4, last lines: “The objective of this study was to examine the effects of the COVID-19 lockdown on diet adherence and stress levels in patients undergoing treatment for laryngopharyngeal reflux (LPR).”

  • Provide more details on methods: Expand the description of the treatment protocol to include specific details about the duration of the treatment period, the dietary modifications implemented, and the behavioral changes recommended to the patients. This will provide a clearer understanding of the interventions employed.

The treatment duration was 3 months.

We specified for medication in methods, p.6, treatment paragraph: “According to the HEMII-pH findings of reflux, patients benefited from a 3-month treatment combining antireflux diet, proton pump inhibitors (PPIs; pantoprazole 20mg once daily), alginate (Gaviscon® 3/d, Reckitt Benckiser, Slough, UK) or magaldrate (Riopan® 3/d, Takeda, Zaventem, Belgium).(6) Patients with acid reflux benefited from pantoprazole (20MG, once daily, fasting) and post-meal alginate (thrice daily), while those with nonacid reflux were treated with post-meal magaldrate or alginate only. Individuals with weakly acid reflux received a combination of pantoprazole (20MG, once daily, fasting) and thrice daily post-meal alginate or magaldrate. Patients with nighttime reflux at the HEMII-pH tracing benefited from an additional alginate or magaldrate (alkaline LPR) at bedtime.(6)

The recommended diet was standardized and validated in two recent publications. See: doi: 10.1007/s00405-019-05631-1 and doi: 10.1007/s00405-019-05711-2.

According to the comment of the reviewer, we provided the specific diet in Appendices 1 and 2 and modified the method section for treatment:

Methods, p.6, Treatment, paragraph 2: “At the first consultation, patients were invited to specify ‘refluxogenic’ foods and beverages that they commonly consumed through a predefined list (10). The antireflux diet was standardized (10), and based on the reduction of foods and beverages associated with a high risk of reflux (10), and the consumption of high-protein, low-fat, alkaline, plant-based foods and beverages (6,10). The list of foods and beverages, which are recommended or discouraged according to these documented impact on gastroesophageal function is provided in Appendices 1 and 2.”

  • Explain the outcomes: Provide a brief explanation of what the reflux symptom score-12 (RSS-12) and reflux sign assessment (RSA) measures entail. This will help readers better understand the clinical features being assessed.

We developed the RSS-12 and RSA paragraph and provided both instrument in Figures 1 and 2. Please see Figures 1 and 2 and the methods: p.6, clinical outcomes, line 1: “Symptoms were evaluated with reflux symptom score-12 (RSS-12),(7)which is a validated 12-item patient reported-outcome questionnaire including otolaryngological, digestive and respiratory symptoms of reflux (Figure 1). The total score ranged from 0 (no symptom) to 300 (frequent and severe symptoms). Reflux sign assessment (RSA) is a validated 61-item clinical instrument, which was developed to rate oral, pharyngeal and laryngeal findings associated with LPR throughout treatment period (8). RSA rates the signs associated with LPR, such as laryngeal erythema or edema (Figure 2).”

  • Include specific result values: Instead of using vague percentages, provide specific result values or ranges. For example, specify the exact percentage of patients who experienced an increase in stress levels or the percentage who reported improved adherence to the antireflux diet during the lockdown.

Done in the abstract as above-mentioned.

  • Provide more specific conclusion: Instead of stating that diet habits improved or remained unchanged in most LPR cases, provide the exact percentage of patients in each category. For example: "Approximately 70% of LPR patients showed improved or unchanged diet habits during the lockdown period."
  • Discuss limitations: Acknowledge any limitations of the study. For instance, mention the small sample size, potential biases in self-reporting, or any other limitations that may impact the generalizability of the findings.

Done in the abstract as above-mentioned.

  • Elaborate on statistical analysis: Provide additional information about the statistical analysis conducted. Include effect sizes, confidence intervals, or other relevant statistical measures to provide a clearer understanding of the relationships found between variables.

By addressing these suggestions, the abstract will become more informative, clearer in its presentation of the study, and provide a better understanding of the research conducted.

 Done in the abstract as above-mentioned.

Introduction

The introduction provides some background information on laryngopharyngeal reflux (LPR) and its association with diet and stress. However, there are few suggestions for improvements that can be addressed:

  • The introduction briefly describes LPR as an inflammatory condition of the upper aerodigestive tract tissues related to gastroduodenal content reflux. It would be helpful to provide a more comprehensive definition or explanation of LPR, including its symptoms, causes, and prevalence.

We added in the introduction: p.4, line 3: “LPR may concern 10% to 30% of outpatients consulting in otolaryngology department (1). The most common symptoms include throat clearing, globus sensation, throat pain, cough, while gastroesophageal reflux disease (GERD) typical symptoms, such as heartburn or regurgitation, are often lacking (1,2).”

  • The introduction states that the consumption of high-fat, high-quick-release sugar, and low-protein foods and beverages, along with stress, may negatively impact the esophageal sphincter tonicity and lead to pharyngeal reflux events. However, it does not explain how these factors specifically contribute to LPR. Providing a more detailed explanation of the relationship between diet, stress, and LPR would enhance the understanding.

These findings are provided in ref. 1 (state of the art review), 2, and 3.

We developed this information in the introduction as requested by the reviewer: Introduction, p.4, line 3: “LPR may concern 10% to 30% of outpatients consulting in otolaryngology department (1). The most common symptoms include throat clearing, globus sensation, throat pain, cough, while gastroesophageal reflux disease (GERD) typical symptoms, such as heartburn or regurgitation, are often lacking (1,2). The consumption of high-fat, high- quick-release sugar, and low-protein foods and beverages and the stress (autonomic nerve dysfunction) are both factors that may negatively influence the esophageal sphincter tonicity, leading to pharyngeal reflux events (1,2). It has been suggested that high-fat, high- quick-release sugar, and low-protein foods may reduce the lower and upper esophageal sphincter tonicities, increase the numbers of transient sphincter relaxation and related gaseous reflux events, last the gastric emptying time, and reduce the up to down motility of esophagus (1). Regarding autonomic nerve function, LPR was associated with an imbalance between sympathetic and para-sympathetic nerve functions, decreasing the para-sympathetic function, which was associated with esophageal sphincter and body dysfunction (1,2).

  • The introduction mentions that the recent COVID-19 pandemic and subsequent lockdowns may have influenced individual lifestyle and diet habits, but it does not specify how or why this impact would occur. Providing a brief explanation of the potential mechanisms or reasons for the impact of lockdown on diet adherence and stress would improve clarity.

We added this information as requested: Introduction, p.4, last paragraph: “Indeed, some patients decided to improve their cooking habits with natural and healthy products, while others increased their consumption of alcohol, snack and fast-food to decrease the stress related to the pandemic (3,4).”

Suggestions for improvements:

  • Provide a comprehensive definition of LPR: Expand the description of LPR to include its symptoms, causes, prevalence, and any relevant statistics or research findings. This will provide a clearer understanding of the condition being studied.
  • Clarify the relationship between diet and LPR: Explain in more detail how the consumption of high-fat, high-quick-release sugar, and low-protein foods and beverages, as well as stress, specifically contribute to the development or exacerbation of LPR. Provide relevant research evidence or theories to support these claims.
  • Explain the potential impact of lockdown: Elaborate on how and why the lockdown measures implemented during the COVID-19 pandemic could influence individual lifestyle and diet habits. Discuss the potential psychological, social, or environmental factors that may contribute to changes in diet adherence and stress levels during lockdown.

By addressing these suggestions, the introduction will become more informative, clearer in its presentation of the research topic, and provide a better understanding of the context and relevance of the study.

Cf above, all were addressed.

Material and Methods

The methods section provides information on the study design, patient selection, diagnostic procedures, treatment protocol, and data analysis. Overall, the section is reasonably clear, but there are a few issues which needs to be addressed:

  • Clarify patient selection criteria: The methods mention that patients with a positive LPR diagnosis at the hypopharyngeal-esophageal multichannel intraluminal impedance-pH monitoring (HEMII-pH) were included in the study. However, it does not specify the specific criteria used for diagnosing LPR or the inclusion criteria for patient selection. Providing more details on these criteria would enhance the understanding of the patient population.

The diagnostic criteria considered: 1) LPR symptoms and findings, 2) the confirmation of diagnosis at the HEMII-pH. >1 pharyngeal reflux event was suggested in a normative data review as the most reliable diagnostic criteria (ref. 5).

We detailed in methods: p.6, line 1: “Patients with laryngopharyngeal symptoms, findings, and a positive LPR diagnosis at the 24-hour hypopharyngeal-esophageal multichannel intraluminal impedance pH-monitoring (HEMII-pH) prior to the COVID-19 lockdown were followed throughout the lockdown period (March to December 2020) in the Department of Otolaryngology–Head & Neck Surgery of CHU Saint-Pierre (Brussels, Belgium). The LPR diagnosis was based on the occurrence of LPR symptoms and >1 acid, weakly acid, or nonacid pharyngeal reflux evens at the HEMII-pH (OFF medication) (5).”

HEMII-pH paragraph: p.6, line 1: “According to a recent systematic review providing normative data for HEMII-pH, the LPR diagnosis criteria was based on the occurrence of >1 acid (pH≤4.0), weakly acid (pH=4.0-7.0), or nonacid (pH>7.0) hypopharyngeal reflux events (off proton pump inhibitors) (5).

  • Provide more details on the treatment protocol: While the treatment approach is briefly described, it would be helpful to provide more information on the duration and frequency of treatment, as well as specific instructions given to patients regarding dietary modifications and medication use. This will help readers better understand the treatment protocol implemented in the study.

We detailed the following information: Methods, treatment, p.8, line 1: According to the HEMII-pH findings of reflux, patients benefited from a 3-month treatment combining antireflux diet, proton pump inhibitors (PPIs; pantoprazole 20mg once daily), alginate (Gaviscon® 3/d, Reckitt Benckiser, Slough, UK) or magaldrate (Riopan® 3/d, Takeda, Zaventem, Belgium).(6) Patients with acid reflux benefited from pantoprazole (20MG, once daily, fasting) and post-meal alginate (thrice daily), while those with nonacid reflux were treated with post-meal magaldrate or alginate only. Individuals with weakly acid reflux received a combination of pantoprazole (20MG, once daily, fasting) and thrice daily post-meal alginate or magaldrate. Patients with nighttime reflux at the HEMII-pH tracing benefited from an additional alginate or magaldrate (alkaline LPR) at bedtime.(6)

At the first consultation, patients were invited to specify ‘refluxogenic’ foods and beverages that they commonly consumed through a predefined list (10). The antireflux diet was standardized (10), and based on the reduction of foods and beverages associated with a high risk of reflux (10), and the consumption of high-protein, low-fat, alkaline, plant-based foods and beverages (6,10). The list of foods and beverages, which are recommended or discouraged according to these documented impact on gastroesophageal function is provided in Appendices 1 and 2.

  • Clarify the evaluation of diet adherence and stress level: The methods mention that patients were invited to evaluate diet adherence and stress levels using a patient-reported outcome questionnaire. However, it would be useful to provide more details on the content and format of the questionnaire. Additionally, clarify whether the evaluation was performed at multiple time points during the follow-up period or only once at the end of the lockdown.

The evaluation was only performed at baseline and after 3-month of treatment. There were no multiple time points.

The influence of lockdown on stress and diet was evaluated with a patient-reported outcome questionnaire, which is available in Appendix 3. Please, see the appendix 3. We did not include this one as figure because we already have 3 tables and 3 figures.

We mentioned this information in the Methods: p.9, lockdown evaluations, line 1: “Patients were invited to evaluate the influence of the lockdown on both the diet adherence and stress level through a short patient-reported outcome questionnaire at baseline and at 3-month posttreatment (Appendix 3).

About the evolution of diet, we specified in Statistical paragraph: p.9, Stat, line 2: “Wilcoxon rank test was used to evaluate the evolution of RSS-12, RSA and PSS from baseline to 3-month posttreatment. The consumption of foods and beverages (weekly versus daily versus no) was assessed with chi-square.”

      While the statistical software used (SPSS) is mentioned, it would be helpful to provide more details on the specific statistical tests employed. Additionally, mention whether any adjustments for multiple comparisons were made.

Cf above. There was no sample size calculation regarding the unpredictable number of patients included from pre- to lockdown period; the lockdown being unpredictable. Methods, statistics, line 4: “There was no sample size calculation.”

We mentioned this point as a limitation in the discussion. Discussion, p.15, last paragraph: “However, the study has some limitations. The low number of patients and the lack of objective testing of autonomic nerve dysfunction (e.g. heart rate variability device) were the primary limitations. The lack of calculation of an effect size is an additional limitation. However, it was difficult to include more patients regarding the short and unpredictable period of study (lockdown periods) and the need to include patients with an objective LPR diagnosis (HEMII-pH). The lack of evaluation of stress during the pre-lockdown period is an additional limitation.

  • Consider adding information on sample size calculation: It would be beneficial to include information on how the sample size was determined. Providing a rationale for the chosen sample size or any power calculations performed would enhance the methodological transparency of the study.

Cf above.

  • The methods mention previous publications and validated questionnaires but do not provide the corresponding references. It is essential to include proper citations to support the use of these tools and ensure transparency.

Cf above.

The references related to RSS-12 (7), RSA (8), and PSS (9) were provided.

By addressing these suggestions, the methods section will become more informative, clearer in its presentation of the study procedures, and provide a better understanding of the research methodology employed.

Results

The results section presents the findings of the study regarding patient characteristics, clinical outcomes, and the influence of the lockdown on diet adherence and stress levels. Overall, the section is clear and provides relevant information. However, there are few suggestions for improvements that can be addressed:

  • Provide specific statistical results: While the results mention that RSS-12 and RSA scores significantly improved from baseline to 3-month posttreatment, it does not provide the actual mean values or effect sizes. Including these specific results will enhance the understanding of the magnitude of improvement.

The mean (SD) and p-value were provided: Results, p.10, line 3: “The mean RSS-12 significantly improved from baseline (66.6 ± 49.1) to 3-month posttreatment (47.6 ± 39.2; p=0.008). The mean pre-treatment RSA (24.2 ± 11.2) significantly improved at 3-month posttreatment (20.3±9.5; p=0.031).”

  • Clarify responder rates: The results mention that 25 patients reported a significant symptom reduction and were considered responders, but it does not specify the criteria used to determine responder status. Providing details on the threshold or criteria for response classification would improve clarity.

The criteria for therapeutic response were detailed in Methods: p.9, treatment, last line: “The medications were titrated at this time regarding the 3-month RSS-12 considering a reduction of >20% of the baseline RSS-12 as an adequate therapeutic response.

In results, p.10, line 5: “Twenty-five patients (78.1%) reported significant symptom reduction (>20% reduction of baseline RSS-12) at posttreatment time and were considered as responders (Table 1).

  • Clarify stress level interpretation: The results mention that the mean PSS at the end of the lockdown corresponded to a high stress level based on normative data. However, it would be helpful to include the actual normative data or references to support this interpretation.

The normative data were detailed in Methods, clinical outcomes, last line: “The normative data reported that a PSS<12 was normal (9). A PSS score between 12 and 21 consists of patients who have stress but adequately managed (mild stress). The stress is moderately managed when the score ranges from 21 to 26 (moderate stress). PSS >26 corresponds to an inadequately managed stress (severe stress).(9)

  • Clarify the influence of lockdown on stress: The results mention that 34.4% of patients believed that the lockdown increased their stress level, while 56.3% reported no influence. However, it does not provide information on the factors contributing to increased stress during the lockdown.

The factors were the pandemic and the lockdown. We rewrote the result section dedicated to stress: Results, p.12: “The mean PSS at the end of the lockdown was 28.3±8.8, which corresponded to high stress level regarding normative data (threshold=12.8±6.2) (8). The stress was related to pandemic and lockdown. Precisely, only one patient reported PSS<13 (3.1%). According to the PSS, 5 (15.6%), 6 (18.8%), and 20 (62.5%) patients reported mild, moderate and severe stress, respectively. Three patients (9.4%) thought that the lockdown period was associated with a better decrease of stress than initially presumed thanks to lockdown period. Eleven patients (34.4%) believed that the lockdown period increased the stress level, while there was no influence of lockdown on stress in 18 patients (56.3%; figure 3B). Overall, 6 patients (18.8%) reported that the lockdown had a negative impact of their LPR. The PSS and RSS-12 scores at the end of the lockdown were significantly correlated (rs=0.681; p<0.001)”

  • Improve the presentation of Table 2: The format and labeling of Table 2 can be improved for better clarity. Consider providing column headers that explicitly indicate the pre-treatment and pandemic periods. Additionally, provide a brief description of the results or observations in the table to aid interpretation.

The table 2 was improved. There were two main column data: Pre-treatment – Pandemic, following by the p-value column. Please, see table 2 in p.13 of results.

Table 2 footnotes: The weakly and daily consumptions of refluxogenic foods and beverages were reported in both pre-treatment and pandemic period. For each period, the number of patients reporting consuming weakly and daily foods or beverages were reported, as well as the sum (N (%)) of weakly and daily refluxogenic foods and beverages. According to Chi-square, the consumption of most refluxogenic foods and beverages significantly decreased from pre-treatment to the lockdown period.  

  • Provide more specific patient characteristics: Instead of just mentioning the mean age and gender distribution, consider including additional relevant patient characteristics such as the range or standard deviation of age, the distribution of age groups, and any other demographic information that may be important for understanding the patient population.

The mean and SD of age and Body Mass Index were provided in Table 1. The age group findings were added in Table 1 as requested by reviewer. Please, see Table 1 in Results, p.10. 

  • Provide specific statistical results: Along with stating that RSS-12 and RSA scores significantly improved, include the actual p-values to indicate the level of significance. Additionally, consider providing effect sizes or confidence intervals to quantify the magnitude of improvement in these scores.

Cf above.

  • Provide references for normative data: When referring to normative data for stress levels, include the specific references or sources from which the normative data was obtained. This will add credibility and allow readers to access the original sources for more information.

Cf above.

By implementing these suggestions, the results section will become more comprehensive, clearer in its presentation of the findings, and provide a better understanding of the study results.

Discussion

The discussion section provides an interpretation of the study findings and highlights the implications of the lockdown on diet habits and stress in patients with LPR. Overall, the section is reasonably clear, but there are a few suggestions for improvements that can be addressed:

  • Lack of clarity in the connection between autonomic nerve dysfunction and pharyngeal reflux events: The discussion mentions that autonomic nerve dysfunction may be characterized by a reduction of vagus nerve activity on the esophagus and sphincters, which increases the risk of pharyngeal reflux events. However, it does not clearly explain how autonomic nerve dysfunction specifically leads to pharyngeal reflux events. Providing a more detailed explanation or referring to relevant studies can improve clarity.

We developed this part of the discussion: p.15, line 3: “Autonomic nerve dysfunction may be characterized by a reduction of the vagus nerve activity on esophagus body and sphincters, which may be associated with an increased number of transient lower and upper esophageal sphincter relaxation episodes, and consequently, the back flow of gastric content into the upper aerodigestive tract. In LPR, pharyngeal reflux events are mainly gaseous and occur post-meals or at distance of the meals (19).”

  • The discussion briefly mentions the potential increase in LPR symptoms related to COVID-19 but does not elaborate on the possible mechanisms or reasons behind this association. Including a brief discussion on the potential impact of COVID-19 on LPR symptoms or any relevant research in this area would enhance the discussion.

We improved this part of the discussion, adding two recent references: p.15, last paragraph: “The findings of the present study were particularly important regarding the potential increase of LPR-symptoms in COVID-19 (21), which may be attributed to the increase of both anxiety and stress in the population. In that way, a recent meta-analysis suggested an increase of dysphonia during and post-COVID-19 infection (22), which should be attributed to LPR. The occurrence of chronic vagus nerve dysfunction in long COVID-19 patients (23) is an additional important issue to explore in future studies.

  • Discuss factors contributing to increased stress: Along with stating the percentage of patients who reported increased stress during the lockdown, discuss potential reasons or factors contributing to this increase. Consider factors such as social isolation, economic concerns, or other relevant aspects that may have influenced stress levels.

We added in the discussion, p.15, last paragraph: “The anxiety and stress were commonly attributed to the pandemic and lockdown. Future studies should be interested to determine the factors underlying the stress and anxiety, which may include social isolation, economic concerns, fear of dying or of the unknown, etc. (24,25).

  • While the limitations of the study are mentioned, there is no explanation provided for how these limitations may have affected the findings or the interpretation of the results. It would be helpful to discuss the potential impact of the low number of patients, the lack of objective testing for autonomic nerve dysfunction, and the lack of evaluation of stress during the pre-lockdown period on the study's conclusions.

We specified, p.16, first lines: “However, the study has some limitations. The low number of patients and the lack of objective testing of autonomic nerve dysfunction (e.g. heart rate variability device) were the primary limitations. The lack of calculation of an effect size is an additional limitation. However, it was difficult to include more patients regarding the short and unpredictable period of study (lockdown periods) and the need to include patients with an objective LPR diagnosis (HEMII-pH). The lack of evaluation of stress during the pre-lockdown period is an additional limitation. The low number of patients may make the findings of the study not generalizable, while it is possible that the level of stress prior to the lockdown was under- or overestimated by patients who evaluated the evolution during the lockdown after potential habituation.”

  • Expand on the importance of stress management: The discussion briefly mentions that the management of stress during the lockdown and pandemic periods is important for LPR patients, but it does not elaborate on why stress management is crucial or how it can impact LPR symptoms. Expanding on the importance of stress management and its potential role in improving LPR symptoms would enhance the discussion, elaborate on the potential mechanisms through which stress can impact LPR symptoms. Discuss how stress may affect esophageal sphincter tone, promote inflammation, or exacerbate autonomic dysfunction. Additionally, highlight the importance of incorporating stress management strategies into the treatment approach for LPR patients and the potential benefits it can have on symptom management.

We added in the discussion: p.15, last lines: “The evolution of stress and anxiety on LPR during lockdown is an important issue according to the burden and cost related to the management of LPR and related complications (e.g. upper aerodigestive tract infections, nasal inflammatory conditions, otitis media, increased risk of vocal cord dysfunction, etc.) (1). The study results may support the need of prevention in potential future similar situation..

  • The conclusion states that the study is the first to describe the influence of lockdown on diet habits and stress in patients with LPR disease. However, this claim should be supported by referencing previous literature or studies that have investigated similar aspects. Providing supporting references would strengthen the conclusion.

We mentioned that this is the first study exploring stress, anxiety and treatment adherence in LPR patients and this is true. There is no similar study in the literature, which is probably related to the lack of pH impedance study availability in many hospitals.

 By addressing these suggestions, the discussion section will provide a more thorough and insightful analysis of the study findings, their implications, and the potential areas for future research.

Reviewer 2 Report

Review for Manuscript ID: medicina-2482639" impact of COVID-19 Lockdown on Laryngopharyngeal Reflux”

The manuscript is not well designed, there are points need to be corrected as follows: 

1.    I think the sample size is very small and a clear conclusion is very difficult to achieve. 

2.    In the abstract: the conclusion is broad, you have to be very specific. 

3.    Introduction is very short and does not support the rationale behind conducting the study. 

4.    References are not according to the journal’s guidelines.

5.    Line 60, what is XX?

6.    Both questionaries used, better to presented as figures.

7.    Figure 1 has no legend. 

8.    In general, the design of the study is very weak with a very small sample size. The rationale behind it is not clear. I think the study design has a major flow. 

BW, 

proofreading is necessary. 

Author Response

Reviewer 2:

The manuscript is not well designed, there are points need to be corrected as follows: 

  1. I think the sample size is very small and a clear conclusion is very difficult to achieve. 

We mentioned the limitations of the study in the discussion of the paper: p.12, last paragraph: “However, the study has some limitations. The low number of patients and the lack of objective testing of autonomic nerve dysfunction (e.g. heart rate variability device) were the primary limitations. The lack of calculation of an effect size is an additional limitation. However, it was difficult to include more patients regarding the short period of study (lockdown periods) and the need to include patients with an objective LPR diagnosis (HEMII-pH). The lack of evaluation of stress during the pre-lockdown period is an additional limitation.”

  1. In the abstract: the conclusion is broad, you have to be very specific. 

We changed: abstract, p.3, last line: “During the lockdown, diet habits of LPR patients were improved in one third and unchanged in 44% of cases. The stress level was increased in one-third of patients, which was associated with an increase of symptom score.”

  1. Introduction is very short and does not support the rationale behind conducting the study.

We improved the introduction: p.5: “Laryngopharyngeal reflux (LPR) is an inflammatory condition of the upper aerodigestive tract tissues related to direct and indirect effect of gastroduodenal content reflux, which induces morphological changes in the upper aerodigestive tract (1). LPR may concern 10% to 30% of outpatients consulting in otolaryngology department (1). The most common symptoms include throat clearing, globus sensation, throat pain, cough, while gastroesophageal reflux disease (GERD) typical symptoms, such as heartburn or regurgitation, are often lacking (1,2). The consumption of high-fat, high- quick-release sugar, and low-protein foods and beverages and the stress (autonomic nerve dysfunction) are both factors that may negatively influence the esophageal sphincter tonicity, leading to pharyngeal reflux events (1,2). It has been suggested that high-fat, high- quick-release sugar, and low-protein foods may reduce the lower and upper esophageal sphincter tonicities, increase the numbers of transient sphincter relaxation and related gaseous reflux events, last the gastric emptying time, and reduce the up to down motility of esophagus (1). Regarding autonomic nerve function, LPR was associated with an imbalance between sympathetic and para-sympathetic nerve functions, decreasing the para-sympathetic function, which was associated with esophageal sphincter and body dysfunction (1,2).

With the recent coronavirus disease 2019 (COVID-19) pandemic, many countries imposed lockdown to reduce the virus spread in the population. Many citizens were confined to home during several weeks, which should influence positively (3) or negatively (4) individual lifestyle and diet habits regarding recent literature. Indeed, some patients decided to improve their cooking habits with natural and healthy products, while others increased their consumption of alcohol, snack and fast-food to decrease the stress related to the pandemic (3,4).

The objective of this study was to examine the effects of the COVID-19 lockdown on diet adherence and stress levels in patients undergoing treatment for laryngopharyngeal reflux (LPR).” 

  1. References are not according to the journal’s guidelines.

We corrected.

  1. Line 60, what is XX?

That was for the blinded assessment of manuscript. Because we are in revision, we however replace the XX by the name of the hospital where the research was conducted.

  1. Both questionaries used, better to presented as figures.

We developed the RSS-12 and RSA paragraph and provided both instrument in Figures 1 and 2. Please see Figures 1 and 2 and the methods: p.6, clinical outcomes, line 1: “Symptoms were evaluated with reflux symptom score-12 (RSS-12),(7) which is a validated 12-item patient reported-outcome questionnaire including otolaryngological, digestive and respiratory symptoms of reflux (Figure 1). The total score ranged from 0 (no symptom) to 300 (frequent and severe symptoms). Reflux sign assessment (RSA) is a validated 61-item clinical instrument, which was developed to rate oral, pharyngeal and laryngeal findings associated with LPR throughout treatment period (8). RSA rates the signs associated with LPR, such as laryngeal erythema or edema (Figure 2).”

  1. Figure 1 has no legend. 

Figure 1 became figure 3. We added legend: p.13: “Figure 3: The figure highlights the influence of lockdown on diet adherence and stress level according to the patient evaluation.”

  1. In general, the design of the study is very weak with a very small sample size. The rationale behind it is not clear. I think the study design has a major flow. 

Yes, we agree but it is difficult in otolaryngology to have true LPR patients (=according to HEMII-pH diagnostic confirmation). This is the reason of the low number of patients. When checking the literature, you can see that the cohorts of LPR patients with a positive diagnosis at the HEMII-pH are small. In our study, the recruitment of patients at key timing regarding pandemic and lockdown complicated the inclusion of a large number of cases. We discussed in the limitation paragraph about this point.

Note that the spelling of the revised paper was checked by a native US speaker.

Round 2

Reviewer 1 Report

Here are my observations on the revised manuscript

Abstract:

·       Adequate

·       All suggested changes were done

Top of Form

Introduction

·       Adequate

·       Significant improvement

·       All suggested changes were done

Material and Methods:

·       Adequate

·       Significant improvement

·       All suggested changes were incorporated in the section

Results:

·       The results section improved

·       Adequate

·       Significant improvement

·       All suggested changes were incorporated in the section

Top of Form

Discussion

·       The discussion section provides an overview of the study findings and compares them to previous research.

·       The results section improved

·       Adequate

·       Significant improvement

·       All suggested changes were incorporated in the section

The manuscript now provides a more comprehensive and informative interpretation of the findings, and provide meaningful insights for future research

Reviewer 2 Report

NIL